# Spatially Explicit Modeling of Anthropogenic Heat Intensity in Beijing Center Area: An Investigation of Driving Factors with Urban Spatial Forms

**DOI:** 10.3390/s23177608

**Published:** 2023-09-01

**Authors:** Meizi Yang, Shisong Cao, Dayu Zhang

**Affiliations:** 1School of Architecture and Urban Planning, Beijing University of Civil Engineering and Architecture, Beijing 100044, China; 1108130321002@stu.bucea.edu.cn; 2School of Geomatics and Urban Spatial Informatics, Beijing University of Civil Engineering and Architecture, Beijing 100044, China; caoshisong@bucea.edu.cn

**Keywords:** anthropogenic heat flux, land function zones, urban spatial form, linear fitting

## Abstract

The escalation of anthropogenic heat emissions poses a significant threat to the urban thermal environment as cities continue to develop. However, the impact of urban spatial form on anthropogenic heat flux (AHF) in different urban functional zones (UFZ) has received limited attention. In this study, we employed the energy inventory method and remotely sensed technology to estimate AHF in Beijing’s central area and utilized the random forest algorithm for UFZ classification. Subsequently, linear fitting models were developed to analyze the relationship between AHF and urban spatial form indicators across diverse UFZ. The results show that the overall accuracy of the classification was determined to be 87.2%, with a Kappa coefficient of 0.8377, indicating a high level of agreement with the actual situation. The business/commercial zone exhibited the highest average AHF value of 33.13 W m^−2^ and the maximum AHF value of 338.07 W m^−2^ among the six land functional zones, indicating that business and commercial areas are the primary sources of anthropogenic heat emissions. The findings reveal substantial variations in the influence of urban spatial form on AHF across different UFZ. Consequently, distinct spatial form control requirements and tailored design strategies are essential for each UFZ. This research highlights the significance of considering urban spatial form in mitigating anthropogenic heat emissions and emphasizes the need for customized planning and renewal approaches in diverse UFZ.

## 1. Introduction

In recent decades, cities worldwide, particularly in developing countries, have undergone rapid urbanization and development [1]. This process has not only transformed natural surfaces with artificial structures [2,3], but has also intensified anthropogenic emissions due to increased population density, energy consumption, and industrial activities. Consequently, these factors have had a significant impact on urban climates, exacerbating the heat island effect and contributing to the occurrence of extreme weather events [4,5]. These adverse effects, in turn, negatively affect the residents’ health, air and water quality, as well as the buildings and infrastructure durability [6,7,8]. In response to the escalating urbanization rates, urban development strategies have gradually shifted from uncontrolled expansion to prioritizing high-quality and endogenous growth. Within this evolving context, climate considerations have gained prominence in urban design and planning processes.

Anthropogenic heat emissions play a significant role in the urban thermal environment as a direct external heat source. By reducing anthropogenic heat emissions, it is possible to improve the overall urban thermal conditions [9]. Therefore, mitigating the impact of climate change by addressing anthropogenic heat emissions has become a crucial consideration in urban planning and design. Research on regulating anthropogenic heat fluxes can be categorized into two scales: urban and building. At the urban scale, studies primarily focus on reducing emissions from transportation sources [10,11,12,13], and developing sustainable urban expansion strategies [14,15,16]. Various approaches such as promoting public transportation, implementing cycling infrastructure, and optimizing traffic management systems have been explored to curb transportation-related emissions. At the building scale, there is a substantial body of research dedicated to enhancing energy efficiency. This includes investigating the benefits of green roofs [17,18], optimizing facade designs [19], improving building envelope insulation [20], enhancing ventilation systems [21], optimizing window designs [22], and developing innovative building materials [23]. By examining both urban and building scales, researchers and practitioners can address the complex interplay between anthropogenic heat emissions, urban design, and building performance to achieve more sustainable and thermally comfortable urban environments.

Moreover, there is evidence to suggest that urban spatial form at the neighborhood level can influence transportation patterns [24,25], urban dynamics [26], and microclimates [27]. These factors, in turn, can directly or indirectly impact anthropogenic heat emissions (Figure 1). However, the research exploring the relationship between urban form and anthropogenic heat is currently limited. It is worth noting that different urban functional zones (UFZ) exhibit distinct human activity patterns, energy consumption patterns, and building characteristics. Consequently, anthropogenic heat emissions within different UFZs may be affected differently by urban spatial form. However, there is a significant gap in research in this particular area. Understanding the complex interactions between urban form, land use patterns, and anthropogenic heat emissions is crucial for developing effective strategies to mitigate the urban heat island effect and promote sustainable urban development. Further research is needed to explore the relationships and dynamics between urban form, land use, and anthropogenic heat emissions across different land functional zones to inform urban planning and design decisions.

This research paper focuses on studying the relationship between urban spatial form and anthropogenic heat emissions in Beijing. The study utilizes multi-source data to classify different UFZ and calculates urban spatial form indices. Anthropogenic heat flux (AHF) is estimated to quantify heat emissions. Box plots analyze AHF variability among UFZ, while linear fitting equations assess the sensitivity of AHF to spatial form factors. The findings inform urban planning to mitigate the urban heat island effect and promote sustainable development in Beijing.

## 2. Study Area and Data

### 2.1. Study Area

The study area of this research is the central urban area of Beijing, a prominent city in northern China with a monsoon-influenced humid continental climate (Figure 2). As the capital of the People’s Republic of China, Beijing has undergone rapid urbanization, resulting in significant economic growth and population density. However, this urban development has also led to changes in the urban surface and increased anthropogenic heat emissions, causing urban climate change. Currently, Beijing has transitioned from rapid and disorderly urbanization to focus on quality improvement. Recent regulations and action plans emphasize green development, energy-saving renovations, and improving living quality and environmental restoration. The central urban area, including districts such as Xicheng, Dongcheng, Haidian, Chaoyang, Fengtai, and Shijingshan, concentrates urban construction and population activities, making it a suitable study area. By examining anthropogenic heat flux (AHF) in this region, the research aims to contribute to the understanding of the urban thermal environment and inform strategies for sustainable urban renewal and development in Beijing.

### 2.2. Data and Preprocessing

(1)Remotely sensed data

Both Luojia-1 and Sentinel 2 satellite data were utilized. The Luojia-1 imagery, acquired in October 2018, was obtained from the Resolution Earth Observation System of the Hubei Data and Application Center (http://www.hbeos.org.cn, accessed on October 2018). These data have a spatial resolution of 130 m and were employed for spatial downscaling of the anthropogenic heat flux (AHF). To address oversaturation and overflow issues in the Luojia-1 data, the researchers employed the Normalized Difference Vegetation Index (NDVI) derived from Landsat 8 images. This allowed them to calculate the Vegetation Adjusted NTL Urban Index (VANUI) [28]. The equations of VANUI were expressed as:(1)$${NTL}_{nor}=\frac{NTL-{NTL}_{min}}{{NTL}_{max}-{NTL}_{min}}$$
(2)$${NDVI}_{max}=MAX\left[{NDVI}_{1}{,\;NDVI}_{2}{,\ldots ,NDVI}_{n}\right]$$
(3)$$VANUI=(1\;-{\;NDVI}_{max}{)\;\times \;NTL}_{nor}$$
where *NTL_nor_* is the normalized *NTL* value, *NTL_max_* and *NTL_min_* are maximum and minimum values in the night light image, *NDVI_max_* means the maximum value in the *NDVI* image, and *VANUI* refers to the Vegetation Adjusted NTL Urban Index.

Sentinel-2A satellite data were employed for UFZ classification. By utilizing the data, the study aimed to accurately classify the UFZ within the study area, contributing to a comprehensive analysis of the relationship between urban spatial form, anthropogenic heat emissions, and land use characteristics. The data used were obtained from the Sentinel Open Access Hub (https://sentinel.esa.int/web/sentinel/sentinel-data-access, accessed on 6 February 2021). Level-1C products, which undergo radiation calibration and geometric correction, were utilized for analysis. The specific bands utilized from the Sentinel-2A data were the blue band (B2), the green band (B3), the red band (B4), and the short-wave infrared (B10). These bands provide valuable information for land classification purposes. The spatial resolution of B2, B3, and B4 is 10 m, while B10 has a spatial resolution of 20 m.

(2)Statistical Data

The statistical data used in this research were sourced from the Statistical Yearbook 2019 for each district in Beijing, which is available through the website of the Beijing Municipal Bureau of Statistics (http://tjj.beijing.gov.cn/, accessed on 2019). These data encompassed various factors, such as land area, resident population, the gross annual value of different industries, energy consumption, and car ownership. These statistical indicators were utilized to calculate the anthropogenic heat flux (AHF) at the district level. By incorporating data on land area, population, industrial output, energy consumption, and other relevant factors, we were able to estimate the magnitude of anthropogenic heat emissions within each district.

(3)Point of Interest (POI) data

A total of 963,374 POI records for Beijing in 2018 were obtained through the Application Programming Interface (API) interface of the Gaode developer platform. The first step is to cleanse the POI data. This process involves eliminating repetitive data, removing incomplete data, and relocating coordinates. Following extraction and cleansing, 554,503 POI data points that were valid were used for further analysis. To classify the POIs based on their functionality, we referred to the “Urban Land Use Classification and Development Land Use Planning Standards (GB 50137-2011)” issued by the Ministry of Housing and Urban-Rural Development of China. The POIs were categorized into six primary functional categories, namely administrative and public services (A), business and commercial facilities (B), green spaces (G), industrial (M), residential (R), and public transportation (S). Table 1 provides a breakdown of the types and numbers of POIs corresponding to each functional category. This classification of POIs allows for a more detailed analysis of the urban spatial form and its relationship to anthropogenic heat emissions in Beijing.

(4)Mobile Signaling data

The cell phone signaling data was collected at specific time points on 8 October 2022, and 22 February 2022, at 2:00, 15:00, and 19:00. Each data entry includes user ID, date, location grid number, latitude, longitude of the location grid center, and stopping point number. China Unicom, with over 100 million subscribers in 2022, provides high sample coverage and is thus representative of the population distribution in Beijing. The data have been desensitized in accordance with the Chinese government’s policy on personal cell phone data control, ensuring that privacy is protected and the data are solely used for scientific research purposes.

(5)Building Data

The building data were obtained from the Urban Data Party, available at the website (https://www.udparty.com/, accessed on December 2018). This dataset provides information on building footprints and urban floor data. The building data were utilized to calculate key urban spatial form indices, including building density (BD), average building height (BH), and floor area ratio (FAR) and building volume (BV). These indices provide insights into the intensity of urban development, vertical growth, and land use efficiency within the study area.

## 3. Method

The research framework of this study, as depicted in Figure 3, comprises three main steps. These steps provide a structured approach in order to analyze the relationship between UFZ and AHF while considering the influence of urban spatial form.

In the first step, the classification of UFZ was accomplished by integrating Sentinel-2A data, point of interest (POI) data, and cellular signal data. This multi-data approach allows for a comprehensive and accurate categorization of different UFZ within the study area.

In the second step, we estimated the annual AHF at the grid level. This estimation was performed using a combination of statistical data and Luojia-1 satellite data. By integrating these datasets, we can quantify the magnitude of anthropogenic heat emissions across the study area.

In the third step, statistical analysis methods were applied to explore the relationship between UFZ and AHF. This analysis aimed to examine how urban spatial form influences the anthropogenic heat flux under different land functional zones. By employing statistical techniques, we seemed to identify significant patterns, correlations, and trends in the data.

Further details and elaboration on the methodology will be presented in subsequent sections of the study, providing a more comprehensive understanding of the research approach and findings.

### 3.1. Urban Functional Zone Classification

In this study, we mapped the UFZ of the central area of Beijing using high-resolution remote sensing data, POIs, and demographic data. The specific three steps were spatial unit generation, feature extraction, and UFZ classification using random forest algorithms.

(1)Spatial unit generation

In previous studies, researchers commonly relied on road networks to define city blocks and utilized these blocks as spatial units for analysis [29,30]. However, in a vast area like the main urban area of Beijing, ensuring the accuracy of road networks can be challenging. Moreover, city blocks often cover large areas, resulting in the coexistence of multiple land functions within a single block. These factors contribute to the reduced accuracy when using blocks as spatial units in the study.

To overcome these challenges and improve accuracy, we proposed the use of a grid system with a resolution of 500 m as the spatial unit. By employing a grid-based approach, we achieved a more detailed representation of the UFZ within the main urban area of Beijing. This approach allows for a finer delineation of different land functions and better captures the heterogeneity of land use patterns in the study area. Additionally, it provides a more accurate representation of the complexities and variations in land use within the study area, leading to more robust and insightful findings.

(2)Feature extraction

A comprehensive set of features was utilized to characterize each spatial unit, incorporating information from spectral data, point of interest (POI) data, and temporal population density. This approach is based on previous research findings [30,31] and aims to capture the diverse aspects of land use and human activity within each spatial unit. The features used are outlined in Table 2.

(1)Spectral features:

Distinct spectral characteristics observe the varying abilities of different features for receiving and emitting electromagnetic waves. Vegetation and built-up areas can be distinguished from one another by utilizing these exclusive spectral characteristics. Therefore, we calculated the spectral features for each parcel within the near-infrared (NIR) range. In this study, the spectral data were taken from various bands of Sentinel-2A satellite data, including the blue, green, red, and short-wave infrared bands. Additionally, typical spectral indices, including the Normalized Vegetation Index (NDVI), that offer insights into vegetation health and coverage were added.

(2)POI Features [29,32,33] and Temporal Population Density:

As the POI data and human activity data are more advantageous in the identification of urban construction land functions, this paper also uses POI data and mobile phone signaling data to identify urban land functions.

For POI features, firstly, we comprised various aspects of the POI data within each spatial unit. Secondly, we calculated the total number of all POIs indicating the overall intensity of points of interest in the area. Thirdly, we calculated the total number, proportion, and term frequency–inverse document frequency (TF-IDF) for each type of POI, capturing the distribution and importance of specific POI categories.

The equations for calculating TF-IDF are expressed as:(4)$${tf}_{i,j}=\frac{n_{i,j}}{\sum_{k}n_{i,j}}$$
(5)$${idf}_{i}=lg\frac{\left|D\right|}{\left|\left\{{j:t}_{i}\in d_{j}\right\}\right|}$$
(6)$${tfidf}_{i,j}{=tf}_{i,j}{\;\times \;idf}_{i}$$
where *i* denotes word, *j* denotes document, *n_i,j_* means the number of occurrences of the word *i* in the document j, and Σ*k n_i,j_* represents the sum of all occurrences of words in the document *j*; |*D*| refers to the total number of documents, and *j:t_i_∈d_j_* is the total number of documents that contain the word *i*. In this study, documents represent spatial units, and words are POIs in parcels.

For temporal population density, it was derived from cellular signal data, which provides an estimation of population distribution and density within each spatial unit. Also, population density values were extracted at different time periods, allowing for an analysis of temporal variations in population distribution and activity patterns.

(3)Random forest classification method

The random forest classifier is a powerful machine-learning algorithm introduced by Leo Breiman and Adele Cutler in 2001 [34,35,36]. It has been extensively studied and proven to exhibit high prediction accuracy, robustness against outliers and noise, and the ability to overcome overfitting tendencies of individual decision trees. The classifier combines the principles of bootstrap aggregating (bagging) and the random subspace method to construct an ensemble of decision trees for classification, regression, and other tasks. The algorithm operates by creating multiple decision trees during the training phase. The specific steps are as follows. First, random forest employs the bootstrap method to randomly select multiple samples with replacement from the original training data. Then, for each bootstrap sample, a decision tree is constructed using the Classification and Regression Tree (CART) algorithm. The selection of optimal internal node branches is based on minimizing the Gini coefficient, a measure of impurity. Finally, the prediction results are obtained through voting, where each decision tree contributes to the final prediction. A 500 m × 500 m grid was adopted as the classification unit, and the random forest algorithm was implemented using the SPSS PRO platform. Through experiments, it was determined that when the number of trees (N) in the random forest ensemble is equal to or greater than 100, the out-of-bag error of each feature type stabilizes, indicating the reliability of the model’s predictions.

(4)Precision evaluation index

The accuracy of the random forest classification for urban functional zones was evaluated using a validation sample. The confusion matrix was employed as a tool to assess the classification accuracy. Several evaluation indexes were calculated based on the confusion matrix, including producer accuracy (PA), user accuracy (UA), overall accuracy (OA), and Kappa coefficient (K). By analyzing the confusion matrix and calculating these indexes, we can evaluate the performance of the classification model and validate its effectiveness in accurately classifying different land use categories in the study area.

### 3.2. Annual AHF Estimation

(1)Estimation of AHF at administrative district level

The calculation of AHFs for each administrative district in Beijing utilized an energy-consumption inventory. The statistical data from the statistical yearbook provided the necessary information for this calculation. Anthropogenic heat emissions were considered to include the energy emissions related to human metabolism, industries, transportation, and buildings, encompassing both commercial and residential structures [14,37,38]. The equation is expressed as:(7)$${AHF}_{y}{=Q}_{m}\;+\;Q_{i}\;+\;Q_{t}\;+\;Q_{b}$$
where *AHF_y_* is the annual anthropogenic heat flux at the district level (W·m^−2^), *Q_m_* is the human metabolic heat flux (W·m^−2^), *Q_i_* is the industry heat flux (W·m^−2^), *Q_t_* is the transportation heat flux (W·m^−2^), and *Q_b_* is the buildings heat flux (W·m^−2^).

The human metabolic heat flux, which represents the energy emitted by humans through their metabolism, can be calculated using the following equation [39]:(8)$$Q_{M}=\frac{\left(P_{1}\;\times \;t_{1}+P_{2}\;\times \;t_{2}\right)\;\times \;N}{\left(t_{1}\;+\;t_{2}\right)\;\times \;A\;\times \;T}$$
where:*P*_1_ is the metabolic rate of sleep state (70 W·person^−1^)*t*_1_ refers to the hours of sleeping time (8 h)*P*_2_ is the metabolic rate of active state (171 W·person^−1^)*t*_2_ is the hours of active time (16 h)*N* is the population*A* is the land area (m^2^)*T* is the duration of the time period considered (1 year).

Industrial anthropogenic heat emissions are mainly derived from various types of energy consumption (e.g., coal, oil, gas, electricity, etc.). In addition, industrial energy consumption needs to be converted into standard coal heat. The equation of the industrial anthropogenic heat flux considers the energy consumption from various sources (such as coal, oil, gas, and electricity) and their respective heat conversion coefficients:(9)$$Q_{I}=\frac{E_{I}\;\times \;C}{A\;\times \;T}$$
where:*E_I_* is the energy consumption of the industry (ton of standard coal equivalent, TCE)*C* refers to the standard coal heat conversion factor (29,306 kJ kg^−1^)*A* is the area of the administrative district (m^2^)*T* is the duration of the time period considered (1 year)

The transportation energy consumption is multiplied by the standard coal heat conversion factor for transportation to convert it into the equivalent heat energy. It is then multiplied by the area of the administrative district and the duration of the time period to estimate the transportation anthropogenic heat flux [38]
(10)$$Q_{t}=\frac{\left(1-\varepsilon_{p}\right)n\;\times \;\bar{\;d\;}{\times \;L\;\times \;m\;\times \;C}_{p}}{A\;\times \;T}$$
where:*ε_p_* is gasoline utilization efficiency (30%)*n* is the sum of civil vehicle$\bar{d}$ is the annual average driving distance per vehicle (11,424.5 km, from Beijing transportation institute)*L* is the fuel consumption per 100 km (12.7 L)*m* is the mass of gasoline per liter (725 g)*C_p_* is the net heat combustion (45 KJ·g^−1^)

Building heat emission is derived from the energy consumption from commercial buildings and residential buildings [39]. The equation of the building heat flux can be expressed as:(11)$$Q_{B}=\frac{\left(E_{BR}\;+\;E_{BC}\right)\;\times \;C}{A\;\times \;T}$$
where:*E_BR_* is the energy consumption of residential buildings (W·m^−2^)*E_BC_* is the energy consumption of commercial buildings (W·m^−2^)

(2)Spatial Downscaling

Extensive research has consistently shown a strong correlation between anthropogenic heat fluxes and nighttime light [38,39,40], prompting the utilization of VANUI for precise geographic allocation of AHF in this study. The process can be summarized as follows: Firstly, the zonal statistics tool in ArcGIS was employed to derive VANUI values for each district. Secondly, a linear regression analysis was conducted to establish an estimation model between the mean AHF of each district, calculated using the energy inventory method, and VANUI values for each district. Finally, based on the estimation model, a refined AHF mapping of Beijing was generated at a spatial resolution of 500 m.

### 3.3. Selection and Calculation of Urban Form Indicators

As shown in Table 3, this study used spatial form, land function, and environment as primary indicators and eight factors, including building density, as secondary indicators. The data are calculated and analyzed in 500 m × 500 m spatial units, and all data are then normalized.

### 3.4. Statistical Analysis

(1)Multiple linear regression

To examine the relationship between AHF and spatial form in different UFZ, this study employed multiple linear regression to construct fitting models. The models aimed to capture the associations between AHF and various spatial form indicators. The linear regression analysis was conducted using the SPSSPRO, and the specific steps were shown in Figure 4.

The study was initiated with a Pearson’s correlation analysis that excluded non-significant variables. As urban form is an influencing factor for AHF, rather than being an extinct factor, we retained correlations with a significant level at both the 0.01 and 0.05 (two-tailed) levels in the correlation analysis. It is also important to note that correlation coefficients obtained from the analysis are not regarded as a precise quantitative relationship between the independent and dependent variables. While correlation analysis is typically employed to describe and present the correlation between variables, it does not indicate the direction of such interactions.

Next, we tested for the presence of covariance between independent variables. In this study, two indicators, the correlation coefficient and the variance inflation factor (VIF), were chosen to measure the severity of multicollinearity. If the correlation coefficient between the two independent variables is greater than 0.8, or VIF is greater than 10, it can be determined that there is a covariance between the two independent variables, and certain measures need to be taken to adjust the independent variables.

Finally, we constructed the multiple regression model and model testing on SPSSPRO. We used the entry fitting method when there was no covariance between the variables, and the stepwise fitting method when there was. To test the model, we used the t-statistic for the regression coefficients and the F-statistic for the regression equations to determine whether or not they were significant at the 0.05 level.

(2)Hotspot analysis

Hotspot analysis is to calculate the Getis-Ord*G*_i_* of each element in the dataset, and the Z-value of the obtained results can reflect the position of the low values clustered in the space, and the *p*-value can reflect the position of the high values clustered in the space. Therefore, the hotspot analysis can be used to understand the distribution trend of cold and hot spots in AHF in central area of Beijing. The equation of Getis-Ord*G*_i_* is:(12)$$G_{i}^{*}=\frac{\sum_{j=1}^{n}W_{ij}X_{j}-\bar{X}\sum_{j=1}^{n}W_{ij}}{S\sqrt{\frac{[n\sum_{j=1}^{n}W_{ij}^{2}-{\left(\sum_{j=1}^{n}W_{ij}^{2}\right)}^{2}]}{n-1}}}$$
where:*X_j_* is the value of the attribute of element *j**W_ij_* is the spatial weights between elements *i* and *j**n* is the total number of the elements

This study used ArcGIS 10.7 for hotspot analysis.

## 4. Result

### 4.1. UFZ Identification Result

#### 4.1.1. Spatial Distribution of the UFZ

Table 4 presents the area and proportion of each UFZ in the study area. Green space comprised the largest portion, covering 450 km^2^ or 34.71% of the total area. Business/commercial functional zones and administrative/public service functional zones followed, accounting for 20.67% and 14.71% of the total area, respectively. Figure 5 illustrates the spatial distribution of the UFZ. As shown, administrative and public service, business and commercial, residential, and transport areas were evenly distributed in the central region, while green spaces were predominantly located in the western side and industrial areas in the eastern side.

#### 4.1.2. Accuracy Assessment

The accuracy of UFZ classification results in this study was assessed using the confusion matrix method. The key accuracy indicators, including user accuracy, producer accuracy, overall accuracy, and Kappa coefficient, were employed. A total of 180 points were randomly selected for validation, and the results are presented in Table 5. The evaluation outcomes indicate that user accuracy for functional zones A and M, as well as producer accuracy for functional zones S and M, were relatively lower. However, the user accuracy and producer accuracy for the remaining UFZ classifications were above 80%. The overall accuracy of the classification was determined to be 87.2%, with a Kappa coefficient of 0.8377, indicating a high level of agreement with the actual situation.

### 4.2. AHF Estimation Result

To better illustrate the spatial distribution of AHF, the AHF was downscaled at precision levels of 130 m and 500 m. The 130 m precision level was used to present the spatial distribution of AHF, while the 500 m precision level was used to correlate the AHF with various urban form indicators.

The results of the AHF spatial distribution at a resolution of 130 m can be obtained. Figure 6 illustrates the presence of significant spatial heterogeneity in AHF. The areas with high AHF values were primarily concentrated between the East 2nd and 4th ring roads, as well as along Chang’an Street. Within this region, numerous grids exhibited AHF values exceeding 500 W/m^2^, which was significantly higher than the average value of 25.27 W/m^2^ in the central urban area of Beijing. It was observed that this area was predominantly occupied by large shopping malls and office buildings, including the Chaoyang CBD. Conversely, the low AHF value areas were mainly found on the outskirts of the central area, particularly in the western flat mountainous region. In these areas, the anthropogenic heat flux measured less than 0.01 W/m^2^. The region was largely covered by natural forests and experiences minimal human development or utilization.

### 4.3. The Differences in Anthropogenic Heat Flux across Different Urban Functional Zone

#### 4.3.1. The Impact of Urban Functions on Anthropogenic Heat Emissions

Figure 7 provides insights into the variations in AHF across different land functional zones. The business/commercial zone exhibited the highest average AHF value of 33.13 W m^−2^ and the maximum AHF value of 338.07 W m^−2^ among the six urban functional zones, indicating that business and commercial areas are the primary sources of anthropogenic heat emissions. Additionally, the residential zone showed significant anthropogenic heat emissions, with a higher mean value of 32.56 W m^−2^ compared to the other zones. This highlights the importance of latent and sensible heat released from energy consumption in residential and commercial buildings, as well as human activities within business/commercial and residential zones, as significant contributors to anthropogenic heat emissions in the main urban area of Beijing.

In contrast to business/commercial zones, industrial and green spaces exhibited relatively lower maximum (82.01 and 105.52 W m^−2^), median (5.03 and 6.57 W m^−2^), and average (8.30 and 8.65 W m^−2^) values of AHF. This implies that the contribution of anthropogenic heat emissions from industrial zones and green spaces is relatively insignificant compared to other land functional zones. While there is limited research specifically examining the characteristics of AHF in different UFZs, studies have indicated that industrial activities can have a significant impact on the urban heat island effect [41,42,43]. The reasons for these different research findings will be further explored and discussed in the subsequent sections.

Furthermore, Figure 7 reveals that business/commercial and public transport zones exhibit high standard deviations for AHF. This suggests significant variations in AHF within these two land functional zones. The observed high standard deviations highlight the heterogeneity in anthropogenic heat emissions within these zones, indicating the presence of localized hotspots or areas with higher energy consumption and heat generation. These variations may be attributed to factors such as the density and intensity of commercial activities, transportation infrastructure, and the distribution of population and buildings within these zones.

#### 4.3.2. The Relationship between UFZ and AHF Hot Spot/Cold Spot

The Getis-Ord-Gi* tool in ArcGIS was employed to conduct a hotspot analysis of AHF, and the proportions of different UFZs within the hotspots and cold spots of AHF were calculated. Figure 8a visually presents the distribution of AHF hotspots and cold spots, revealing a concentration of AHF hotspots in the central part of the city, while cold spots were predominantly observed in the western and eastern areas. Figure 8b displays the proportions of each land functional zone within the cold spots area, with Zone G exhibiting the highest percentage (62.1%) of land function in the cold spots, followed by Zones M (27.0%), B (5.2%), A (3.4%), S (1.7%), and R (0.6%). Figure 8c illustrates the proportions of UFZ within the hotspot areas, where Zone B had the highest proportion (35.3%), trailed by Zones R (34.1%), A (22.7%), G (5.3%), M (2.1%), and S (0.5%). These findings highlight the prevalence of green space in areas characterized by low AHF values, whereas business/commercial, residential, and administrative/public service functional zones dominate in regions with high AHF values. A comparison between Figure 8b,c further indicates that green space predominantly occupies the cold spot areas, with minimal representation of other land functions. Conversely, no single land functional zone dominates in the hotspot areas, with similar proportions of business/commercial, residential, and administrative/public service functional zones observed.

### 4.4. The Influence of Urban Spatial Forms on AHF under Varying UFZ

Based on the screening of the influencing factors and the refinement of the quantification methodology above, this section targets the quantitative analysis of the mechanism of AHF influencing factors in different UFZs from the aspect of urban form. To further explore the influence of different urban forms within various UFZ on AHF, we performed linear regression analysis between AHF and three types of urban form indicators within each UFZ. To minimize errors, we used the stratified mean of AHF for the analysis and normalized the data to account for differences in dimensionality among the indicators.

Correlation analysis was employed for the initial screening of urban form factors. The outcomes indicate that AHFs in various UFZs are impacted by urban form factors, and there are variations in the level of influence of the factors in distinct UFZs. (Appendix A). To be specific, in functional zone A, B, and R, AHF and FC were not correlated, so FC should be excluded from the regression model; in functional zone S, AHF, and BH, FD were not correlated, so BH and FD should be excluded from the regression model; and in functional zones G and M, AHF was correlated with all the morphology indices, so all the indices should be retained.

The results of correlation analyses can only describe the relationship between the independent variables and the dependent variable, but cannot determine whether all of these independent variables can be included in the regression model. If multicollinearity exists between the independent variables, it reduces the stability and accuracy of the regression model. The results of the multicollinearity test show that in all types of sites, although the VIF values of all independent variables are less than 10, but in functional zones M, G, and S, FAR and BD there is a high correlation, and the Person correlation coefficient is greater than 0.8 (Appendix A). To put it differently, there may exist an issue of multicollinearity between the two independent variables. Hence, to construct the regression models for these three types of functional zones in future studies, the stepwise regression method is essential, while the regression models for the other three zones are constructed using the entry method.

After testing (Appendix A), we can see that the regression model was statistically significant (*p* < 0.05). The R^2^ of the regression model ranged from 0.2 to 0.7. Considering that the urban form factor is only an influencing factor of the AHF and not a determining factor, the study considers the explanatory power of the model, i.e., the R^2^, to be acceptable. Table 6 shows the results of the regression model test for each type of land use.

The normalized coefficients represent the effect on the dependent variable due to the change of independent variable per unit. The percentage of standardized coefficients for factor are used to compare the contributions of different urban form factors on AHF models (Appendix A).

In functional zone A and S, the contribution of a unit change of FVC was larger than other factors, indicating that the FVC plays the most important role in these functional zones. As for functional zone B, R, and M, FAR is the most important factor. As for functional zone G, AIA is the most important factor.

## 5. Discussion

### 5.1. Factors Influencing the Variability of AHF across Different UFZ

#### 5.1.1. Uneven Economic Development Can Contribute to Heterogeneity in AHF

Contrary to previous studies [42,43], our findings reveal an interesting phenomenon of relatively lower AHF in industrial areas compared to other functional zones. This can be attributed to the fact that because of the policy of prohibiting of new construction and expansion of factories in central Beijing, and relocation of highly polluting and energy-intensive factories, many factories in these areas have become idle due to inadequate development in line with the urbanization process, and some of the remaining sites were converted into industrial theme parks. Other factories have undergone an industrial transformation to relatively low-energy-consuming industries, such as art and design, product R&D, and technology services. Taking Chaoyang District as an example, it served as an industrial base for Beijing, hosting major industries such as textiles, electronics, chemicals, machinery, and automobiles since the 1950s. By the late 1950s, more than 60 large and medium-sized industrial enterprises had established themselves in Chaoyang District. However, with rapid urbanization, some industrial enterprises started relocating to restore the environmental quality of the capital, leaving numerous factories unused. Consequently, these areas experienced a lack of economic and social vitality, leading to extremely low anthropogenic heat fluxes. Other industrial enterprises, such as VINTAGE (Figure 9a), the Beijing Music Industrial Park (Figure 9b), have undertaken industrial upgrading and transformation, significantly reducing energy consumption, environmental pollution, and heat emissions.

Furthermore, the variability of AHF within the same UFZ can be further illustrated by the influence of human activity and economic development. Taking business/commercial zones as an example, we observe that well-operated commercial areas with high foot traffic exhibited higher AHF values. Prominent examples include Guomao Shopping Mall, Xidan Shopping Center, and Wangfujing Shopping Center, with AHF values exceeding 500 W/m^2^. Conversely, inefficient buildings with poor commercial operations demonstrate lower anthropogenic heat flux. For instance, buildings such as Jingxi Science and Technology Building in Shijingshan and Xinqi Wang Office Building in Dongcheng had AHF values below 20 W/m^2^. Some small commercial buildings even fall below 5 W/m^2^, which was lower than the average AHF value of 33.13 W/m^2^ for business/commercial zones.

From the above analysis, it is evident that social and economic developments currently result in significant anthropogenic heat emissions, and it is challenging to separate them. These findings highlight the intertwined nature of social and economic factors in contributing to AHF variations.

#### 5.1.2. The Spatial Form Characteristic of Different UFZ Contribute to Heterogeneity in AHF

AHF exhibits heterogeneity across different UFZ, which can be attributed to the distinct urban spatial forms within each zone. Administrative/Public Service areas are characterized by predominantly open mid-rise or open low-rise buildings with moderate building densities. These areas often feature buildings between 9 and 18 m in height and are accompanied by extensive green spaces or activity areas. Residential areas primarily consist of open mid-rise and high-rise buildings, with some dense low-rise structures in historic districts. Residential zones have specific requirements for daylighting and ventilation, resulting in mandatory regulations regarding building height, density, floor area ratio (FAR), layout, and landscaping. Business/commercial areas are characterized by dense high-rise structures, reflecting higher building densities and FAR. Industrial areas typically comprise large, dense, low-rise structures with minimal surrounding vegetation, primarily serving industrial purposes. Green spaces are predominantly composed of vegetation and water bodies, with only a few scattered buildings in some areas. Public transport buildings are generally large, low-rise structures. These differences in spatial form across UFZ contribute to the variations in AHF, highlighting the influence of urban design and land use on AHF distribution.

These differences in spatial form may contribute to variations in AHF observed across different UFZs. Figure 10 also demonstrates that functional zones B, A, and R had higher building densities, heights, and FAR, while zones M, G, and S exhibited lower values. This pattern aligns with the statistics for AHF emissions in different UFZ presented in Figure 6, further indicating a correlation between spatial characteristics and UFZ.

### 5.2. The Effect of Urban Form on AHF Varies between Different UFZ

Previous studies have primarily focused on the overall influence of spatial form on AHF or the urban heat island effect, overlooking the impact of spatial form within different locations [42,44]. In our study, we specifically address this gap by examining different UFZ, which has led to the discovery of several interesting phenomena.

Firstly, based on the previous preliminary identification of the influencing factors and the regression model between urban form elements and AHF composed of different types of UFZ, we can see that the two factors of FVC and BV appear in all the equations, and at the same time, FVC is the most influential urban form indicator of functional zones A and S. Therefore, great attention should be paid to the control of these two indicators in urban planning and design. On the other hand, FC has almost no effect on AHF in most of the UFZs and only a weak impact on AHF in functional zone M. Therefore, FC may not be essential in urban planning and design.

Secondly, we can see that the influence of urban form elements on AHF varies due to the change in urban functions. Therefore, proposing specific spatial optimization strategies for different UFZs based on the varying sensitivity of AHF to the urban form is essential. For instance, in functional zones A, M, and G, increasing the building height increases AHF, whereas in functional zones R and B, decreases AHF. And another example, in functional zones A and S, they are highly influenced by the FVC, so sufficient green spaces should be preserved. However, in functional zone M, B, and R, they are highly influenced by the FAR, so the super-high-intensity developments should be avoided. In functional zone S, an integrated vertical development model involving both surface and underground development can be adopted.

Finally, our study reveals that building density does not have a significant effect on anthropogenic heat emissions in most UFZs, which differs from the findings of Chen et al. [44]. This discrepancy can be attributed to the different climatic zones of the study regions. Our research focuses on Beijing, situated in the warm temperate zone, characterized by colder and drier temperatures. In contrast, Chen et al. [44] examined Shanghai, Nanjing, and Hangzhou in the subtropical zone, which experience a stronger demand for ventilation and cooling in summer. In Beijing’s colder climate, higher building density contributes to reduced heat loss, resulting in higher temperatures [45,46]. Consequently, higher building density can somewhat alleviate the demand for heating during winter, leading to lower anthropogenic heat emissions. On the other hand, in the Yangtze River Delta region, characterized by higher humidity and temperature, relatively taller buildings with lower density facilitate air circulation, reducing humidity and temperature [45]. This can result in a decreased demand for air conditioning and cooling in the Yangtze River Delta.

These findings emphasize that the impact of urban form on anthropogenic heat varies significantly across different climatic zones, and even between urban centers and suburban areas. As a result, the optimization of urban form should be tailored to each region. Northern regions and suburban areas may benefit from high-density multi-story spatial forms, which can help reduce heating demand. Conversely, southern regions and urban centers may be better suited for low-density high-rise spatial forms, which promote air circulation and can mitigate cooling needs, especially in humid climates like the Yangtze River Delta.

In conclusion, our study emphasizes the heterogeneity of spatial form’s impact on AHF across UFZs. It underscores the importance of considering the specific characteristics of each UFZ and climatic zone when designing urban environments.

## 6. Conclusions

This study aims to investigate the relationship between anthropogenic heat flux and urban spatial form within different land function zones and derive optimal development strategies to mitigate anthropogenic heat emissions in each UFZ. Based on the findings of this study, the following conclusions can be drawn:

(1) Business/commercial functional zones exhibited the highest AHF among all UFZ, while residential and administrative/public service functional zones also had relatively high AHF levels. In contrast, green spaces demonstrated the lowest AHF, significantly lower than the other zones. Therefore, special attention should be given to the protection of existing parks in Beijing’s urban center, and urban sprawl towards the west and north, where significant ecological green spaces exist, should be avoided.

(2) The floor area ratio and fractional vegetation cover strongly influences the AHF in all UFZs, emphasizing the importance of controlling land development intensity and protecting of existing green spaces.

(3) The impact of urban form elements on AHF varies across urban functional zones. The AHF of administrative/public functional and street/transportation zones was highly influenced by the fractional vegetation cover; the AHF of business, residential, and industrial functional zones was highly influenced by the floor area ratio; and the green space functional zones was highly influenced by the impervious surface area. So, we need to develop different strategies in different urban functional zones

By considering these conclusions, urban planners and policymakers can develop targeted strategies to mitigate anthropogenic heat emissions and optimize urban development in different UFZ. These strategies will contribute to creating more sustainable, livable, and environmentally friendly urban environments.

## Figures and Tables

**Figure 1 sensors-23-07608-f001:**
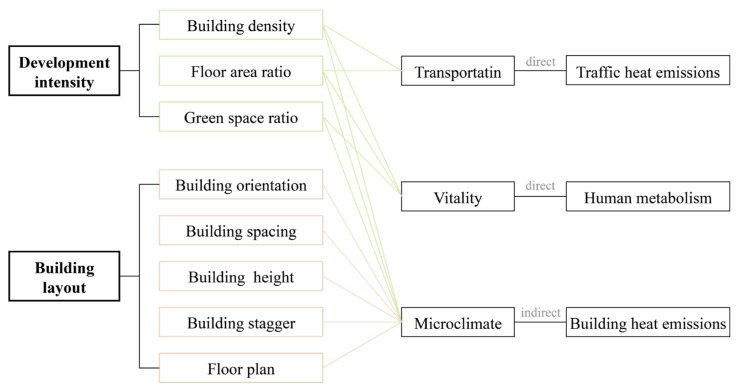
Factors influencing anthropogenic heat emissions from neighborhoods-level urban spatial form.

**Figure 2 sensors-23-07608-f002:**
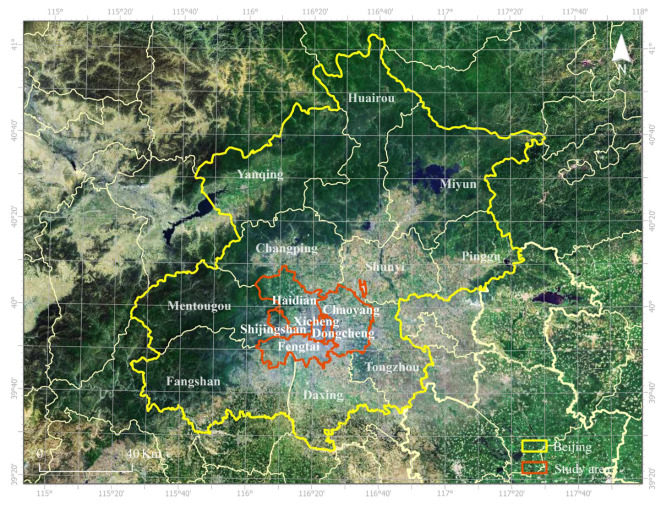
Location of the core area of Beijing.

**Figure 3 sensors-23-07608-f003:**
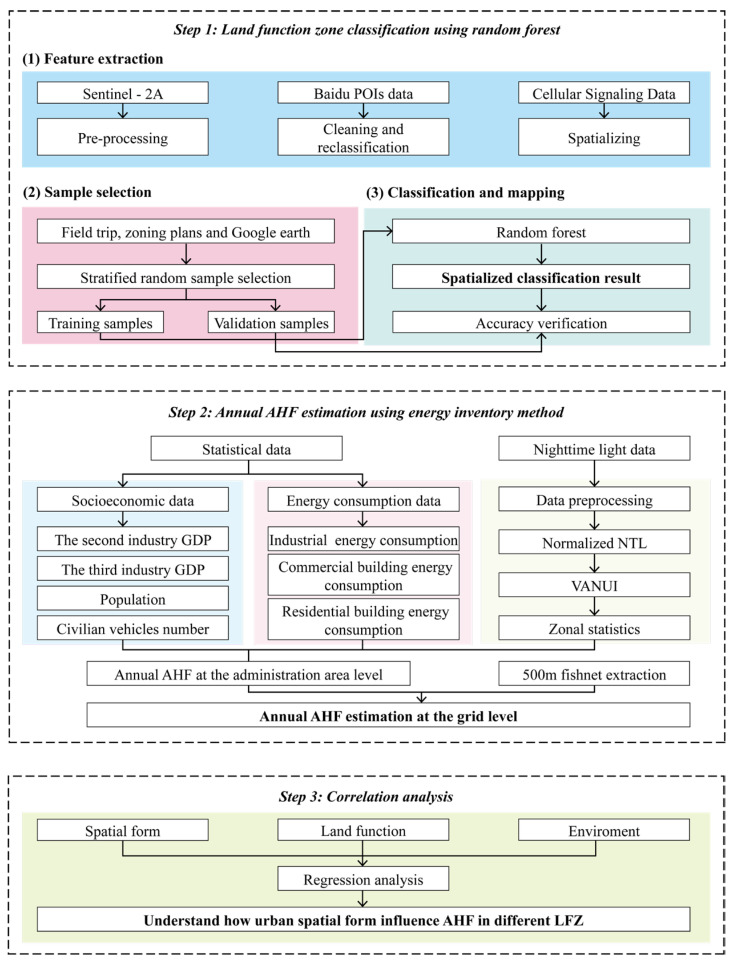
The overall research framework diagram.

**Figure 4 sensors-23-07608-f004:**
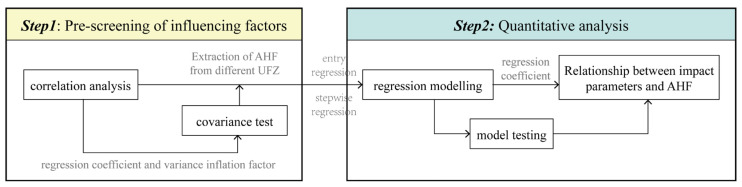
Statistical analysis framework.

**Figure 5 sensors-23-07608-f005:**
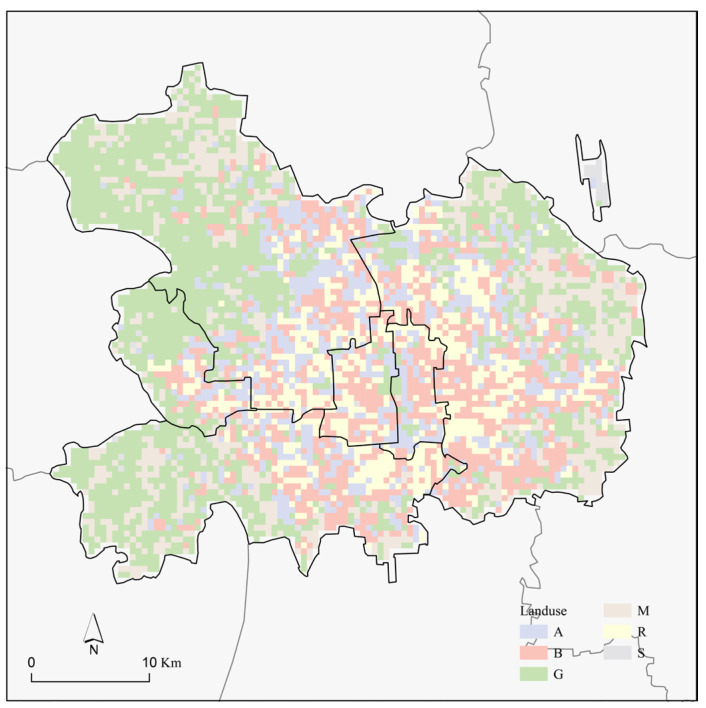
Spatial distribution of urban functional zones.

**Figure 6 sensors-23-07608-f006:**
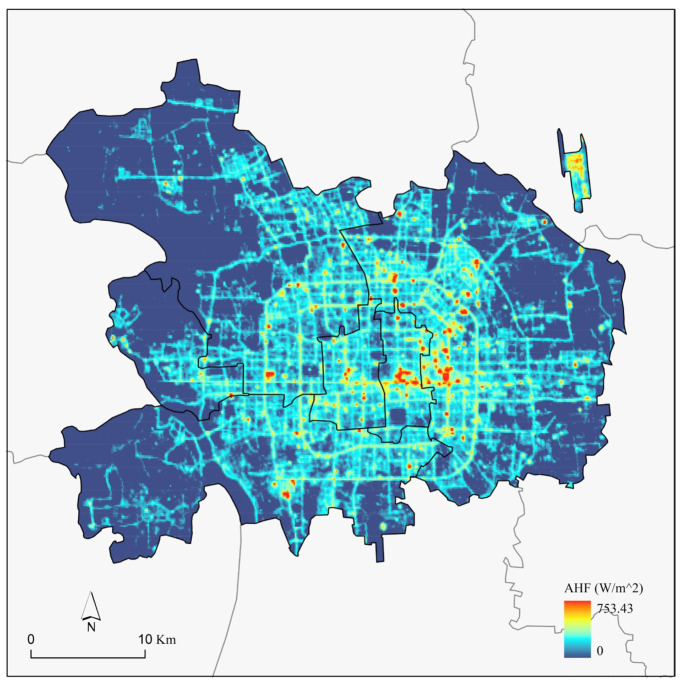
Spatial distribution of anthropogenic heat flux.

**Figure 7 sensors-23-07608-f007:**
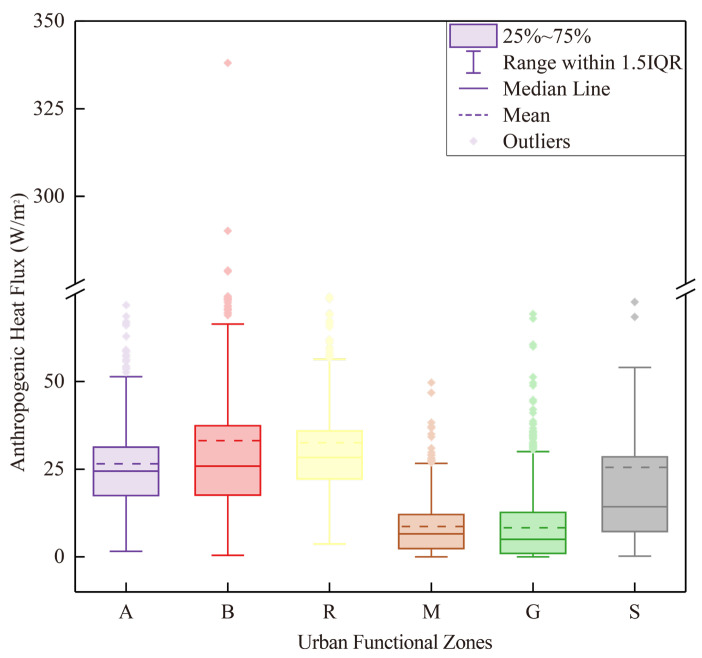
Spatial statistics of anthropogenic heat flux and different urban functional zones.

**Figure 8 sensors-23-07608-f008:**
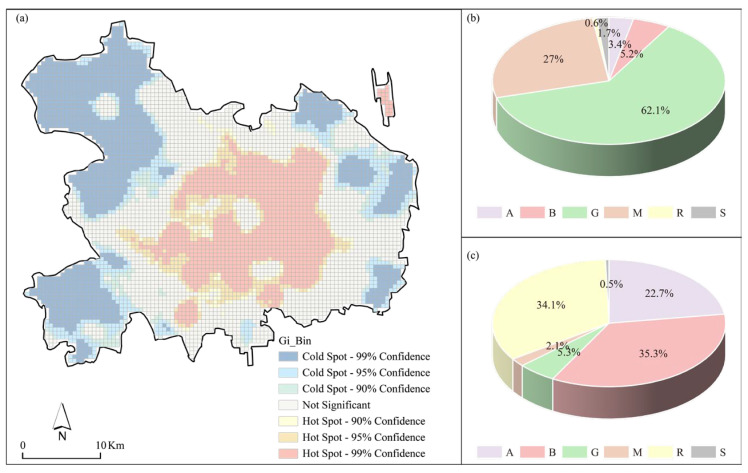
Proportions of different urban functional zones within the cold spots and hotspots. (**a**) Hotspot analysis of anthropogenic heat fluxes; (**b**) urban functional zones in cold spots; and (**c**) urban functional zones in hotspots.

**Figure 9 sensors-23-07608-f009:**
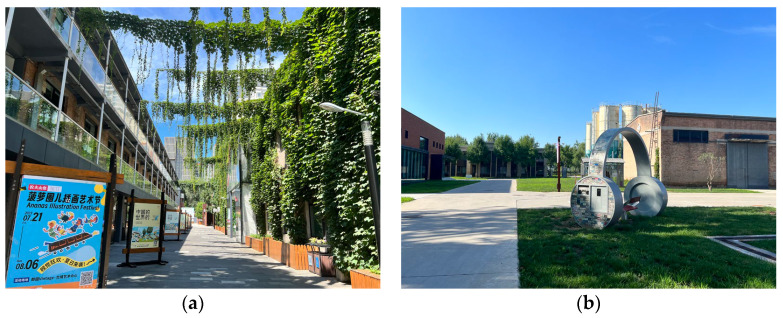
Typical industrial functional zones in central area of Beijing. (**a**) VINTAGE; (**b**) Beijing Music Industrial Park.

**Figure 10 sensors-23-07608-f010:**
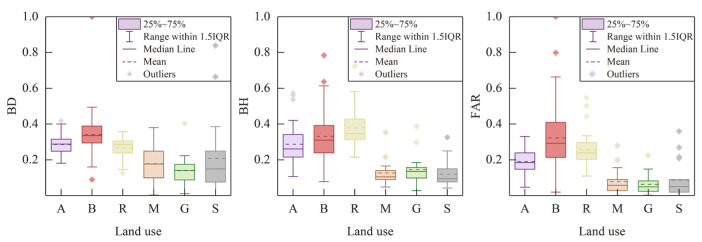
Spatial form differences between different urban functional zones.

**Table 1 sensors-23-07608-t001:** Land-use and POI classification system.

Urban Function Category	POI Type	Numbers	Proportion
Residential (R)	Residential; Residential Related	26,761	4.83%
Business and Commercial Facilities (B)	Catering Service; Shopping Service; Accommodation Service; Financial and Insurance Services; Life Service;	408,725	73.71%
Administration and Public Service (A)	Science, Education and Culture; Sports and Leisure; Medical Care; Government Agencies and Social Organizations	73,565	13.27%
Industrial (M)	Corporation	7237	1.31%
Green Space (G)	Park and Plaza; Scenic Spot	10,189	1.84%
Street and Transportation (S)	Transportation Service Facilities	28,026	5.05%

**Table 2 sensors-23-07608-t002:** List of features for urban functional zone classification in this study.

Feature Information	Parameter
Spectral	Mean and standard deviation of red, green, blue, near-infrared, and two short-wave infrared bands; Mean NDVI
POI	Total number of all POIs and each type of POIs; The proportion of each type of POIs; The TF-IDF of each type of POIs
Time series population density	Population density values at 2:00, 15:00, and 19:00 on a weekday and weekend

**Table 3 sensors-23-07608-t003:** Urban form indicators.

Urban Form Indicators		Calculation Formula
Primary Indicators	Secondary Indicators	
Spatial Form	BD	BD=FA/A *FA* is totals building floor area, *A* is spatial unit area
	FAR	FAR=BA/A *BA* is totals building area, *A* is spatial unit area
	BH	Average height of buildings in the spatial unit
	BV	Average volume of buildings in the spatial unit
Land Function	FC	$FM=-\sum_{i=1}^{n}(p_{i}\;\times \;{lnp}_{i})$ *i* is the total numbers of the POIs types, *pi* is the ratio of the number of POIs of type *i* to the total number of POIs in the spatial unit
	FD	The total number of POIs in the land unit
Environment	FVC	$\mathrm{FVC}=(\mathrm{NDVI}\;-{\mathrm{NDVI}}_{\mathrm{soil}})/({\mathrm{NDVI}}_{\mathrm{veg}}-{\mathrm{NDVI}}_{\mathrm{soil}})$ *NDVI_veg_* is the NDVI value of pure vegetation, *NDVI_soil_* is the NDVI value of pure bare soil, *NDVI* is the NDVI value of in the spatial unit
	AIA	Surface covered by impermeable materials

**Table 4 sensors-23-07608-t004:** Land area and proportion of each functional zones.

LFZ	Area (km^2^)	Proportion
A	190.75	14.71%
B	268.00	20.67%
R	180.25	13.90%
M	189.00	14.58%
G	450.00	34.71%
S	18.50	1.43%

**Table 5 sensors-23-07608-t005:** The confusion matrix of urban functional land classification.

Reference Data	Classes						
	B	A	R	M	G	S	PA
B	48	2	2	1	0	0	0.91
A	0	33	2	1	2	0	0.87
R	2	6	32	0	0	0	0.80
M	1	1	0	11	1	0	0.79
G	0	0	0	0	28	0	1.00
S	1	0	0	1	0	5	0.71
UA	0.89	0.79	0.89	0.79	0.90	1.00	

**Table 6 sensors-23-07608-t006:** Regression results of urban form indicators.

UFZ	Fitting Equation
A	y = 0.072 − 0.063 × BD + 0.037 × BH + 0.076 × FAR + 0.684 × BV + 0.054 × FD − 0.074 × FVC + 0.028 × AIA
B	y = 0.065 − 0.152 × BD − 0.051 × BH + 0.361 × FAR + 0.328 × BV + 0.157 × FD − 0.054 × FVC + 0.026 × AIA
R	y = 0.126 − 0.313 × BD − 0.1 × BH + 0.167 × FAR + 2.59 × BV + 0.115 × FD −0.097 × FVC + 0.064 × AIA
M	y = 0.031 + 0.027 × BH + 0.101 × FAR-0.031 × FVC + 0.008 × FC
G	y = 0.026 + 0.024 × AIA + 0.033 × BH − 0.027 × FVC + 0.106 × FD + 0.014 × FC + 0.042 × FAR
S	y = 0.181 − 0.239 × FVC + 0.264 × BV

## Data Availability

The data used in this study are available upon reasonable request to the corresponding author.

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
