# Peer review of "Spatially Explicit Modeling of Anthropogenic Heat Intensity in Beijing Center Area: An Investigation of Driving Factors with Urban Spatial Forms"

_sensors, 2023, doi:10.3390/s23177608_

Round 1

Reviewer 1 Report

The manuscript is well-written and may be considered for publication after the following clarifications.

1.      The use of Local Climate Zones (LCZs) [Stewart & Oke, 2012] instead of urban functional zones (UFZ) is more suitable for such kinds of studies and will be helpful in comparing with the other cities. I suggest authors to consider LCZs which present a more reproducible and consistent definition of urban forms.

2.       Methods section is difficult to follow. I suggest authors to provide elaborate details and how different steps are connected with each other. For example: Step 1: Land function zone classification using random forest does not provide information on how datasets from different sources are being utilized in the sample selection.

3.      It is difficult to understand what is the spatial resolution of the AHF and UFZs. In the methods section is mentioned to be 500 x 500m grids, however, in the results section (3.2) it is 130m. Please provide clarifications.

4.      Results: Line 379: It is counterintuitive to have “industrial” have similar values as that of green spaces. I suppose here, “industrial” AH values are not properly represented as it is expected to have more AH emissions due to the chemical processes. I suggest authors have a better representation of the industrial AH. Using only energy consumption might underrepresent industrial AH.  

5.      Section 3.3.2: The methodology for hot/cold spots must be explained in the methods section.

6.      Section 4.1.1: It is highly unexpected to have industrial AH than other functional types, and I suggest authors to provide a detailed explanation for the results. Also, please provide the locations on the map for international readers.

7.      Section 4.3: This is based on mere speculations rather than concrete evidence. I suggest authors either remove this section or provide evidence based on the results of the study. For example: the authors imply that applying different controls on AH emission it will reduce the urban heat island effect. However, no such evidence of reducing urban heat island effect due to such measures is shown in the results section.

8.      Similarly, I suggest either removing or modifying the recommendations in the conclusions section of the manuscript.

Minor suggestions:

1.      Line 83: Authors have worked on “Urban Heat Island”. I suggest reconsidering this statement.

2.      Line 109: “the researchers”. Please cite correctly. For example, Li et al. in 2020 …

3.      Line 140: “cleaning”. Please provide more details on “cleaning” of the dataset.

4.      Table 3: “LFZ”. I guess the authors mean “UFZ”

5.      Figure 4 seems to be missing from the manuscript.

6.      Line 401: Please provide on “Getis-Ord-Gi” tool.

Stewart, I. D., & Oke, T. R. (2012). Local climate zones for urban temperature studies. Bulletin of the American Meteorological Society, 93(12), 1879-1900.

None

Author Response

  1. The use of Local Climate Zones (LCZs) [Stewart & Oke, 2012] instead of urban functional zones (UFZ) is more suitable for such kinds of studies and will be helpful in comparing with the other cities. I suggest authors to consider LCZs which present a more reproducible and consistent definition of urban forms.

LCZs is a very important classification basis, but on the one hand, the classification of LCZs is related to urban form in some extent, so it is difficult to further study the relationship between AHF and urban form under this type of classification. On the other hand, the influence of UFZs on AHF is also important and less scholars have studied on it, so this paper chooses the classification of UFZs.

  1. Methods section is difficult to follow. I suggest authors to provide elaborate details and how different steps are connected with each other. For example: Step 1: Land function zone classification using random forest does not provide information on how datasets from different sources are being utilized in the sample selection.

We have rewritten the methods section.

  1. It is difficult to understand what is the spatial resolution of the AHF and UFZs. In the methods section is mentioned to be 500 x 500m grids, however, in the results section (3.2) it is 130m. Please provide clarifications.

To better illustrate the spatial distribution of AHF, the AHF was downscaled at preci-sion levels of 130m and 500m. The 130m precision level was used to present the spatial distribution of AHF, while the 500m precision level was used to correlate the AHF with various urban form indicators.

  1. Results: Line 379: It is counterintuitive to have “industrial” have similar values as that of green spaces. I suppose here, “industrial” AH values are not properly represented as it is expected to have more AH emissions due to the chemical processes. I suggest authors have a better representation of the industrial AH. Using only energy consumption might underrepresent industrial AH.
  2. Section 4.1.1: It is highly unexpected to have industrial AH than other functional types, and I suggest authors to provide a detailed explanation for the results. Also, please provide the locations on the map for international readers.

Contrary to previous studies, our findings reveal an interesting phenomenon of relatively lower AHF in industrial areas compared to other functional zones. This can be attributed to the fact that because of the policy of prohibiting of new construction and expansion of factories in central Beijing, and relocation of highly polluting and energy-intensive factories, many factories in these areas have become idle due to inadequate development in line with the urbanization process, some of the remaining sites were converted into industrial theme parks. Other factories have undergone an industrial trans-formation to relatively low-energy-consuming industries such as art and design, product R&D and technology services.

The location was shown in figure 2, we also add photographs for the better understanding.

  1. Section 3.3.2: The methodology for hot/cold spots must be explained in the methods section.

We add the explanations of the methodology for hot/cold spots in the methods section.

  1. Section 4.3: This is based on mere speculations rather than concrete evidence. I suggest authors either remove this section or provide evidence based on the results of the study. For example: the authors imply that applying different controls on AH emission it will reduce the urban heat island effect. However, no such evidence of reducing urban heat island effect due to such measures is shown in the results section.
  2. Similarly, I suggest either removing or modifying the recommendations in the conclusions section of the manuscript.

We removed section 4.3 and recommendations in the conclusions section.

Reviewer 2 Report

This paper adopts a various range of datasets and methods for a comprehensive investigation of the relationship between urban spatial form and anthropogenic heat flux (AHF), further clarifying differences in this relationship by urban functional zone (UFZ). The research framework presented in Figure 1 is very interesting, having potential to invite more research in this AHF domain regarding UHI. Methodologically, this paper is novel. My suggestions are regarding Figure 8, the linear modeling of the correlation between urban form and AHF. (1) Instead of this simple bivariate correlation analysis, multiple linear regression is much more appropriate where one can estimate the impact of one urban form variable while the impacts of other forms are controlled for. (2) Where are all the other urban form indicators in this analysis (e.g., building spacing)? A linear regression model would be one of the most efficient and parsimonious ways to analyze the impact of individual urban form variables while curating this result in a concise table/figure. 

Author Response

 (1)  Instead of this simple bivariate correlation analysis, multiple linear regression is much more appropriate where one can estimate the impact of one urban form variable while the impacts of other forms are controlled for.

We changed the simple bivariate correlation analysis to the multiple linear regression.

(2)  Where are all the other urban form indicators in this analysis (e.g., building spacing)? A linear regression model would be one of the most efficient and parsimonious ways to analyze the impact of individual urban form variables while curating this result in a concise table/figure.

We are sorry for the misunderstanding, the urban form indicators in Figure 1 are just summaries for the previous study, not the indicators we meant to use. But, still, we added more indicators to our paper.
